# Continuous Regularized Wasserstein Barycenters

**Lingxiao Li**
MIT CSAIL
lingxiao@mit.edu

**Aude Genevay**
MIT CSAIL
aude.genevay@gmail.com

**Mikhail Yurochkin**
IBM Research, MIT-IBM Watson AI Lab
mikhail.yurochkin@ibm.com

**Justin Solomon**
MIT CSAIL, MIT-IBM Watson AI Lab
jsolomon@mit.edu

## Abstract

Wasserstein barycenters provide a geometrically meaningful way to aggregate probability distributions, built on the theory of optimal transport. They are difficult to compute in practice, however, leading previous work to restrict their supports to finite sets of points. Leveraging a new dual formulation for the regularized Wasserstein barycenter problem, we introduce a stochastic algorithm that constructs a *continuous* approximation of the barycenter. We establish strong duality and use the corresponding primal-dual relationship to parametrize the barycenter implicitly using the dual potentials of regularized transport problems. The resulting problem can be solved with stochastic gradient descent, which yields an efficient online algorithm to approximate the barycenter of continuous distributions given sample access. We demonstrate the effectiveness of our approach and compare against previous work on synthetic examples and real-world applications.

## 1 Introduction

In statistics and machine learning, it is often desirable to aggregate distinct but similar collections of information, represented as probability distributions. For example, when temperature data is missing from one weather station, one can combine the temperature histograms from nearby stations to provide a good estimate for the missing station [Sol+14]. Or, in a Bayesian inference setting, when inference on the full data set is not allowed due to privacy or efficiency reasons, one can distributively gather posterior samples from slices of the data to form a single posterior incorporating information from all the data [Min+14; SLD18; Sri+15; Sta+17].

One successful aggregation strategy consists in computing a *barycenter* of the input distributions. Given a notion of distance between distributions, the barycenter is the distribution that minimizes the sum of distances to the individual input distributions. A popular choice of distance is the *Wasserstein distance* based on the theory of optimal transport. The corresponding barycenter, called the *Wasserstein barycenter* was first studied in [AC11]. Intuitively, the Wasserstein distance is defined as the least amount of work required to transport the mass from one distribution into the other, where the notion of work is measured with respect to the metric of the underlying space on which the distributions are supported. The Wasserstein distance enjoys strong theoretical properties [Vil08; FG15; San15], and efficient algorithms for its computation have been proposed in recent years [Cut13; Gen+16; Seg+17; PC19]. It has found success in many machine learning applications, including Bayesian inference [EM12] and domain adaptation [CFT14].

Finding the Wasserstein barycenter is not an easy task. To make it computationally tractable, the barycenter is typically constrained to be a discrete measure on a fixed number of support points [CD14; Sta+17; Dvu+18; CCS18]. This discrete approximation, however, can be undesirable in downstream

applications, as it goes against the inherently continuous nature of many data distributions and lacks the capability of generating fresh samples when needed. To address this shortcoming, in this work we compute a continuous approximation of the barycenter that provides a stream of samples from the barycenter.

**Contributions.** We propose a stochastic algorithm to approximate the Wasserstein barycenter without discretizing its support. Our method relies on a novel dual formulation of the regularized Wasserstein barycenter problem where the regularization is applied on a *continuous* support measure for the barycenter. The dual potentials that solve this dual problem can be used to recover the optimal transport plan between each input distribution and the barycenter. We solve the dual problem using stochastic gradient descent, yielding an efficient algorithm that only requires sample access to the input distributions. The barycenter can then be extracted as a follow-up step. Compared to existing methods, our algorithm produces the first continuous approximation of the barycenter that allows sample access. We demonstrate the effectiveness of our approach on synthesized examples and on real-world data for subset posterior aggregation.

**Related Work.** In [AC11], the notion of Wasserstein barycenters was first introduced and analyzed theoretically. Although significant progress has been made in developing fast and scalable methods to compute the Wasserstein distance between distributions in both discrete [Cut13] and continuous cases [Gen+16; Seg+17], the search for an efficient and flexible Wasserstein barycenter algorithm has been overlooked in the continuous setting.

To have a tractable representation of the barycenter, previous methods assume that the barycenter is supported on discrete points. When the support is fixed *a priori*, the problem boils down to estimating the weights of the support points, and efficient projection-based methods can be used for discrete input measures [Ben+15; Sol+15; CP16] while gradient-based solvers can be used for continuous input measures [Sta+17; Dvu+18]. These fixed-support methods become prohibitive in higher dimensions, as the number of points required for a reasonable *a priori* discrete support grows exponentially. When the support points are free to move, alternating optimization of the support weights and the support points is typically used to deal with the non-convexity of the problem [CD14]. More recent methods use stochastic optimization [CCS18] or the Franke–Wolfe algorithm [Lui+19] to construct the support iteratively. These free-support methods, however, are computationally expensive and do not scale to a large number of support points.

If the support is no longer constrained to be discrete, a key challenge is to find a suitable representation of the now continuous barycenter, a challenge that is unaddressed in previous work. We draw inspiration from [Gen+16], where the Wasserstein distance between continuous distributions is computed by parameterizing the dual potentials in a reproducing kernel Hilbert space (RKHS). Their work was followed by [Seg+17], where neural networks are used instead of RKHS parameterizations. The primal-dual relationship exhibits a bridge between continuous dual potentials and the transport plans, which can the be marginalized to get a convenient continuous representation of the distributions. However, a direct extension of [Seg+17] to the barycenter problem will need to parameterize the barycenter measure, resulting in an alternating min-max optimization. By introducing a novel regularizing measure that does not rely on the unknown barycenter but only on a proxy support measure, we are able to encode the information of the barycenter in the dual potentials themselves without explicitly parameterizing the barycenter. This idea of computing the barycenter from dual potentials can be viewed as a generalization of [CP16] to the continuous case where the barycenter is no longer supported on a finite set known beforehand.

## 2   Background on Optimal Transport

Throughout, we consider a compact set $\mathcal{X} \subset \mathbb{R}^d$ equipped with a symmetric cost function $c : \mathcal{X} \times \mathcal{X} \to \mathbb{R}_+$. We denote by $\mathcal{M}_1^+(\mathcal{X})$ the space of probability Radon measures. For any $\mu, \nu \in \mathcal{M}_1^+(\mathcal{X})$, the Kantorovich formulation of optimal transport between $\mu$ and $\nu$ is defined as:

$$W(\mu, \nu) \stackrel{\triangle}{=} \inf_{\pi \in \Pi(\mu,\nu)} \int_{\mathcal{X}^2} c(x,y) \, \mathrm{d}\pi(x,y), \tag{1}$$

where $\Pi(\mu, \nu) \stackrel{\triangle}{=} \{\pi \in \mathcal{M}_1^+(\mathcal{X}^2) | (P_x)_\# \pi = \mu, (P_y)_\# \pi = \nu\}$ is the set of admissable *transport plans*, $P_x(x,y) \stackrel{\triangle}{=} x$ and $P_y(x,y) \stackrel{\triangle}{=} y$ are the projections onto the first and second coordinate

respectively, and $T_\#(\alpha)$ denotes the pushforward of the measure $\alpha$ by a function $T$. When $c(x,y) = ||x - y||_2^p$, the quantity $W(\mu, \nu)^{1/p}$ is the *p-Wasserstein* distance between $\mu$ and $\nu$.

The primal problem (1) admits an equivalent dual formulation [San15]:

$$W(\mu, \nu) = \sup_{\substack{f,g \in C(\mathcal{X}) \\ f \oplus g \leq c}} \int_\mathcal{X} f(x)\, \mathrm{d}\mu(x) + \int_\mathcal{X} g(y)\, \mathrm{d}\nu(y), \qquad (2)$$

where $C(\mathcal{X})$ is the space of continuous real-valued functions on $\mathcal{X}$, and $(f \oplus g)(x,y) \overset{\triangle}{=} f(x) + g(y)$. The inequality $f \oplus g \leq c$ is interpreted as $f(x) + g(y) \leq c(x,y)$ for $\mu$-a.e. $x$ and $\nu$-a.e. $y$. We refer to $f$ and $g$ as the *dual potentials*.

Directly solving (1) and (2) is challenging even with discretization, since the resulting linear program can be large. Hence *regularized* optimal transport has emerged as a popular, efficient alternative [Cut13]. Let $\xi \in \mathcal{M}_1^+(\mathcal{X}^2)$ be the measure on which we enforce a relaxed version of the constraint $f \oplus g \leq c$ that we call the *regularizing measure*. In previous work, $\xi$ is usually taken to be the product measure $\mu \otimes \nu$ [Gen+16] or the uniform measure on a discrete set of points [Cut13]. Given a convex regularizer $R : \mathbb{R} \to \mathbb{R}$, we define the regularized version of (1) with respect to $\xi, R$ as

$$W_R^\xi(\mu, \nu) \overset{\triangle}{=} \inf_{\substack{\pi \in \Pi(\mu, \nu) \\ \pi \ll \xi}} \int_{\mathcal{X} \times \mathcal{X}} c(x,y)\, \mathrm{d}\pi(x,y) + \int_{\mathcal{X} \times \mathcal{X}} R\left(\frac{\mathrm{d}\pi}{\mathrm{d}\xi}\right) \mathrm{d}\xi(x,y), \qquad (3)$$

where $\pi \ll \xi$ denotes that $\pi$ is absolutely continuous with respect to $\xi$. In this work, we consider entropic and quadratic regularization defined by

$$\forall t \geq 0, \quad R(t) \overset{\triangle}{=} \begin{cases} \varepsilon(t \ln t - t) & \text{entropic} \\ \frac{\varepsilon}{2} t^2 & \text{quadratic.} \end{cases} \qquad (4)$$

As in the unregularized case, the primal problem (3) admits an equivalent dual formulation for entropic [Gen+16; Cla+19] and quadratic [LMM19] regularization:

$$W_R^\xi(\mu, \nu) = \sup_{f,g \in C(\mathcal{X})} \int_\mathcal{X} f(x)\, \mathrm{d}\mu(x) + \int_\mathcal{X} g(y)\, \mathrm{d}\nu(y) - \int_{\mathcal{X} \times \mathcal{X}} R^* \left(f(x) + g(y) - c(x,y)\right) \mathrm{d}\xi(x,y), \qquad (5)$$

where the regularizer $R^*$ on the dual problem is determined as

$$\forall t \in \mathbb{R}, \quad R^*(t) = \begin{cases} \varepsilon \exp\left(\frac{t}{\varepsilon}\right) & \text{entropic} \\ \frac{1}{2\varepsilon}(t_+)^2 & \text{quadratic.} \end{cases} \qquad (6)$$

The regularized dual problem has the advantage of being *unconstrained* thanks to the penalization of $R^*$ to smoothly enforce $f \oplus g \leq c$. We can recover the optimal transport plan $\pi$ from the optimal dual potentials $(f, g)$ using the primal-dual relationship [Gen+16; LMM19]:

$$\mathrm{d}\pi(x,y) = H(x,y)\mathrm{d}\xi(x,y), \text{ where } H(x,y) = \begin{cases} \exp\left(\frac{f(x)+g(y)-c(x,y)}{\varepsilon}\right) & \text{entropic} \\ \left(\frac{f(x)+g(y)-c(x,y)}{\varepsilon}\right)_+ & \text{quadratic.} \end{cases} \qquad (7)$$

The entropic regularizer is more popular, as in the discrete case it yields a problem that can be solved with the celebrated Sinkhorn algorithm [Cut13]. We will consider both entropic and quadratic regularization in our setup, although more general regularizers can be used.

## 3 Regularized Wasserstein Barycenters

We can now use the regularized Wasserstein distance (3) to define a regularized version of the classic Wasserstein barycenter problem introduced in [AC11].

### 3.1 Primal and dual formulation of the regularized Wasserstein barycenter

Given input distributions $\mu_1, \ldots, \mu_n \in \mathcal{M}_1^+(\mathcal{X})$ and weights $\lambda_1, \ldots, \lambda_n \in \mathbb{R}_+$, the *(unregularized) Wasserstein barycenter* problem[1] of these input measures with respect to the weights is [AC11]:

$$\inf_{\nu \in \mathcal{M}_1^+(\mathcal{X})} \sum_{i=1}^n \lambda_i W(\mu_i, \nu). \tag{8}$$

Since this formulation is hard to solve in practice, we instead consider the following *regularized Wasserstein barycenter* problem with respect to the regularized Wasserstein distance (3) for some $\eta \in \mathcal{M}_1^+(\mathcal{X})$, where $R$ refers to either quadratic or entropic regularization (4):

$$\inf_{\nu \in \mathcal{M}_1^+(\mathcal{X})} \sum_{i=1}^n \lambda_i W_R^{\mu_i \otimes \eta}(\mu_i, \nu). \tag{9}$$

If we knew the true barycenter $\nu$, we would set $\eta = \nu$. Hence for (9) to make sense *a priori*, we must use another measure $\eta$ as a proxy for $\nu$. We call such $\eta$ the *barycenter support measure*. If no information about the barycenter is known beforehand, we take $\eta = \text{Unif}(\mathcal{X})$, the uniform measure on $\mathcal{X}$. Otherwise we can choose $\eta$ based on the information we have.

Our method relies on the following dual formulation of (9):

**Theorem 3.1.** *The dual problem of (9) is*

$$\sup_{\substack{\{(f_i, g_i)\}_{i=1}^n \subset C(\mathcal{X})^2 \\ \sum_{i=1}^n \lambda_i g_i = 0}} \sum_{i=1}^n \lambda_i \left( \int f_i \, \mathrm{d}\mu_i - \iint R^* \left( f_i(x) + g_i(y) - c(x, y) \right) \, \mathrm{d}\mu_i(x) \, \mathrm{d}\eta(y) \right). \tag{10}$$

*Moreover, strong duality holds in the sense that the infimum of* (9) *equals the supremum of* (10), *and a solution to* (9) *exists. If* $\{(f_i, g_i)\}_{i=1}^n$ *solves* (10), *then each* $(f_i, g_i)$ *is a solution to the dual formulation* (5) *of* $W_R^{\mu_i \otimes \eta}(\mu_i, \nu)$.

We include a proof of Theorem 3.1 in the supplementary document. The proof relies on the convex duality theory of locally convex topological spaces as in [ET99].

**Remark 3.1.1.** *Based on Theorem 3.1, we can recover the optimal transport plan* $\pi_i$ *between* $\mu_i$ *and the barycenter* $\nu$ *from the pair* $(f_i, g_i)$ *solving* (10) *via the primal-dual relationship* (7).

### 3.2 Solving the regularized barycenter problem

Notice that (10) is convex in the potentials $\{f_i, g_i\}_{i=1}^n$ with the linear constraint $\sum_{i=1}^n \lambda_i g_i = 0$. To get an unconstrained version of the problem, we replace each $g_i$ with $g_i - \sum_{i=1}^n \lambda_i g_i$. Rewriting integrals as expectations, we obtain the following formulation equivalent to (10):

$$\sup_{\substack{\{f_i\}_{i=1}^n \subset C(\mathcal{X}) \\ \{g_i\}_{i=1}^n \subset C(\mathcal{X})}} \mathbb{E}_{\substack{X_i \sim \mu_i \\ Y \sim \eta}} \left[ \sum_{i=1}^n \lambda_i \left( f_i(X_i) - R^* \left( f_i(X_i) + g_i(Y) - \sum_{j=1}^n \lambda_j g_j(Y) - c(X_i, Y) \right) \right) \right]. \tag{11}$$

This version is an unconstrained concave maximization of an expectation.

The optimization space of (11) is infinite-dimensional. Following [Gen+16], we parameterize the potentials $\{f_i, g_i\}_{i=1}^n$ and solve (11) using stochastic gradient descent (Algorithm 1). In their paper, the parameterization is done using reproducing kernel Hilbert spaces, which can be made more efficient using random Fourier features [RR08]; this technique gives convergence guarantees but is only well-suited for smooth problems. In [Seg+17], a neural network parameterization is used with the benefit of approximating arbitrary continuous functions, but its convergence guarantees are more elusive. We extend these techniques to solve (11). A comparison between neural network parameterization and random Fourier parameterization is included in Figure 2.

**Algorithm 1:** Stochastic gradient descent to solve the regularized barycenter problem (11)

---

**Input :** distributions $\mu_1, \ldots, \mu_n$ with sample access, weights $(\lambda_1, \ldots, \lambda_n)$, dual regularizer $R^*$,
        regularizing measure $\eta$, cost function $c$, gradient update function `ApplyGradient`.

Initialize parameterizations $\{(f_{\theta_i}, g_{\phi_i})\}_{i=1}^n$;
**for** $l \leftarrow 1$ **to** $n_{\text{epochs}}$ **do**
    |  $\forall i \in \{1, \ldots, n\}$: sample $x^{(i)} \sim \mu_i$;    sample $y \sim \eta$;
    |  $\bar{g} \leftarrow \sum_{i=1}^n \lambda_i g_{\phi_i}(y)$;
    |  $F \leftarrow \sum_{i=1}^n \lambda_i \left( f_{\theta_i}(x^{(i)}) - R^* \left( f_{\theta_i}(x^{(i)}) + g_{\phi_i}(y) - \bar{g} - c(x^{(i)}, y) \right) \right)$;
    |  for $i = 1, \ldots, n$: `ApplyGradient`$(F, \theta_i)$;   `ApplyGradient`$(F, \phi_i)$;
**return** dual potentials $\{(f_{\theta_i}, g_{\phi_i})\}_{i=1}^n$.

---

Once we approximate the optimal potentials $\{f_i\}_{i=1}^n, \{g_i\}_{i=1}^n$, as observed in Remark 3.1.1, we can recover the corresponding transport plan $\pi_i$ via the primal-dual relationships (7).

This formulation can be easily extended to the discrete case. If the barycenter has a fixed discrete support known *a priori*, we can take $\eta$ to be the uniform measure on the discrete support and parameterize each $g_i$ as a real-valued vector. If the input distributions are discrete, we can use an analogous discrete representation for each $f_i$.

### 3.3 Recovering the barycenter

Given the optimal transport plan $\pi_i \in \mathcal{M}_1^+(\mathcal{X}^2)$ for each $i$, the barycenter $\nu$ equals $(P_y)_{\#}\pi_i$ for any $i$ by Theorem 3.1. While this pushforward is straightforward to evaluate when $\pi_i$'s are discrete, in the continuous setting such marginalization is difficult, especially when the dimension of $\mathcal{X}$ is large. Below we suggest a few ways to recover the barycenter from the transport plans:

(a) Use numerical integration to approximate $(P_y)_{\#}\pi_i(x) = \int \pi_i(x, y)\,\mathrm{d}y$ with proper discretization of the space $\mathcal{X}$, if $\pi_i$ has density (if the input distributions and $\eta$ have densities by (7)).

(b) Use Markov chain Monte Carlo (MCMC) methods to sample according to $\pi_i$, again assuming it has (unnormalized) density, and then take the second components of all the samples.

Option (a) is only viable for small dimensions. Option (b) is capable of providing quality samples, but is slow in practice and requires case-by-case parameter tuning. Both (a) and (b) additionally require knowing the densities of input distributions to evaluate $\pi_i$, which may not be available in practice.

A different kind of approach is to estimate a *Monge map* approximating each $\pi_i$. Formally, a Monge map from $\mu \in \mathcal{M}_1^+(\mathcal{X})$ to $\nu \in \mathcal{M}_1^+(\mathcal{X})$ is a solution to $\inf_{T:\mathcal{X}\to\mathcal{X}} \int_{\mathcal{X}} c(x, T(x))\mathrm{d}\mu(x)$ such that $T_{\#}(\mu) = \nu$. When the cost satisfies $c(x, y) = h(x - y)$ with a convex $h$ and $\mu$ has density, it is linked to the optimal transport plan $\pi$ between $\mu$ and $\nu$ by $\pi = (\mathrm{id}, T)_{\#}(\mu)$ [San15]. With regularization, such exact correspondence may not hold. Nevertheless $\pi$ encodes the crucial information of a Monge map when the regularization is small. If we can find $T_i : \mathcal{X} \to \mathcal{X}$ that realizes $\pi_i$ for each $i$, then we can recover the barycenter as $\sum_{i=1}^n \lambda_i (T_i)_{\#}\mu_i$. In the unregularized case, all of $(T_i)_{\#}\mu_i$ should agree. In practice, we have found that taking the weighted average of $(T_i)_{\#}\mu_i$'s helps reduce the error brought by each individual $T_i$. We consider the following variants of Monge map estimation:

(c) Compute pointwise barycentric projection [CFT14; Seg+17]. If $c(x, y) = \|x - y\|_2^2$, then barycentric projection takes the simplified form

$$T_i(x) = \mathbb{E}_{Y \sim \pi_i(\cdot|x)} [Y]. \tag{12}$$

(d) Recover an approximation of the Monge map using the gradient of the dual potentials [TJ19]. For the case when $c(x, y) = \|x - y\|_2^2$ and the densities of the source distributions exist, there exists a unique Monge map realizing the (unregularized) optimal transport plan $\pi_i$ [San15]:

$$T_i(x) = x - \frac{1}{2}\nabla f_i(x). \tag{13}$$

While this does not strictly hold for the regularized case, it gives a cheap approximation of $T_i$'s.

(e) Find $T_i$ as a solution to the following optimization problem [Seg+17], where $H$ is defined in (7):

$$T_i \stackrel{\triangle}{=} \underset{T:\mathcal{X}\to\mathcal{X}}{\operatorname{argmin}} \underset{(X,Y)\sim\pi_i}{\mathbb{E}} \left[c(T(X),Y)\right] = \underset{T\in\mathcal{X}\to\mathcal{X}}{\operatorname{argmin}} \underset{\substack{X\sim\mu_i \\ Y\sim\eta}}{\mathbb{E}} \left[c(T(X),Y)H(X,Y)\right]. \qquad (14)$$

In [Seg+17] each $T_i$ is parameterized as a neural network. In practice, the regularity of the neural networks smooths the transport map, avoiding erroneous oscillations due to sampling error in methods like barycentric projection (c) where each $T_i$ is estimated pointwise.

Compared to (a)(b), options (c)(d)(e) do no require knowing the densities of the input distributions. See a comparison of these methods in Figure 1.

## 4   Implementation and Experiments

We tested the proposed framework for computing a continuous approximation of the barycenter on both synthetic and real-world data. In all experiments we use equal weights for input distributions, i.e., $\lambda_i = \frac{1}{n}$ for all $i = 1, \ldots, n$. Throughout we use the squared Euclidean distance as the cost function, i.e., $c(x,y) = \|x-y\|_2^2$. Note that our method is not limited to Euclidean distance costs and can be generalized to different cost functions in $\mathbb{R}^d$—or even to distance functions on curved domains. The source code is publicly available at `https://github.com/lingxiaoli94/CWB`.

**Implementation details.** The support measure $\eta$ is set to be the uniform measure on a box containing the support of all the source distributions, estimated by sampling.

For $c(x,y) = \|x-y\|_2^2$, we can simplify the (unregularized) Wasserstein barycenter problem by considering centered input distributions [Álv+16]. Concretely, if the mean of $\mu_i$ is $m_i$, then the mean of the resulting barycenter is $\sum_{i=1}^n \lambda_i m_i$, and we can first compute the barycenter of input distributions centered at 0 and then translate the barycenter to have the right mean. We adopt this simplification since this allows us to reduce the size of the support measure $\eta$ when the input distributions are far apart. When computing the Monge map (c)(d)(e), for each $i$, we further enforce $(T_i)_{\#}(\mu_i)$ to have zero mean by replacing $T_i$ with $T_i - \mathbb{E}_{X\sim\mu_i}[T_i(X)]$. We have found that empirically this helps reduce the bias coming from regularization when recovering the Monge map.

The stochastic gradient descent used to solve (11) and (14) is implemented in Tensorflow 2.1 [Aba+16]. In all experiments below, we use Adam optimizer [KB14] with learning rate $10^{-4}$ and batch size 4096 or 8192 for the training. The dual potentials $\{f_i, g_i\}_{i=1}^n$ in (11) are each parameterized as neural networks with two fully-connected layers ($d \to 128 \to 256 \to 1$) using ReLU activations. Every $T_i$ in (14) is parameterized with layers ($d \to 128 \to 256 \to d$). We have tested with deeper/wider network architectures but have found no noticeable improvement. We change the choice of the regularizer and the number of training iterations depending on the examples.

**Qualitative results in 2 and 3 dimensions.** Figure 1 shows the results for methods (a)-(e) from section 3.3 on various examples. For each example represented as a row, we first train the dual potentials using quadratic regularization with $\varepsilon = 10^{-4}$ or $\varepsilon = 10^{-5}$. Then each method is run subsequently to obtain the barycenter. Algorithm 1 takes less than 10 minutes to finish for these experiments.[2] For (a) we use a discretized grid with grid size 200 in 2D, and grid size 80 in 3D. For (b) we use Metropolis-Hastings to generate $10^5$ samples with a symmetric Gaussian proposal. The results from (a)(b) are aggregated from all transport plans. For (c)(d)(e) we sample from each input distribution and then push the samples forward using $T_i$'s to have $10^5$ samples in total.

In short: (a) numerical integration shows the transport plans $\pi_i$'s computed by (7) are accurate and smooth; (b) MCMC samples match the barycenter in (a) but are expensive to compute and can be blurry near the boundaries; (c) barycentric projection yields poor boundaries due to the high variance in evaluating (12) pointwise; (d) gradient-based map has fragmented white lines in the interior; (e) the method by [Seg+17] can inherit undesirable artifact from the input distributions—for instance, in the last column of the second row the digit 3 looks pixelated.

Next, we compare the impact of the choice of regularization and parameterization in Figure 2. We use the digit 3 example (row 2 in Figure 1) and run numerical integration (a) to recover the barycenter.

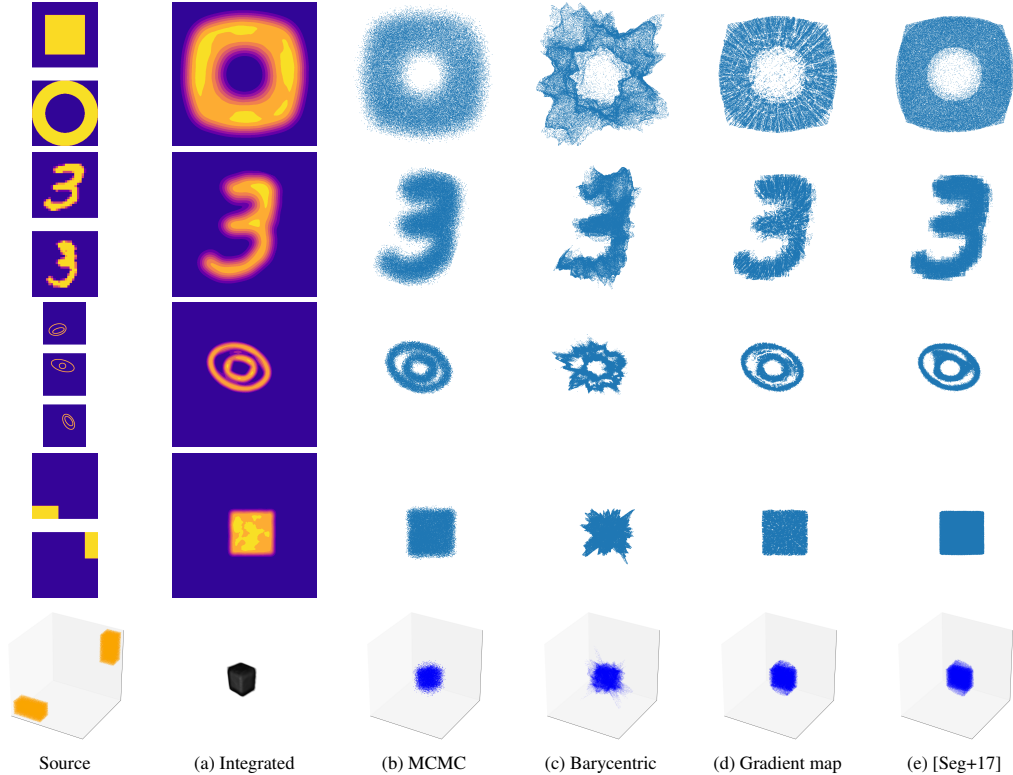

|         |                |          |               |                 |            |
|---------|----------------|----------|---------------|-----------------|------------|
| Source  | (a) Integrated | (b) MCMC | (c) Barycentric | (d) Gradient map | (e) [Seg+17] |

Figure 1: Comparison of barycenter recovery methods.

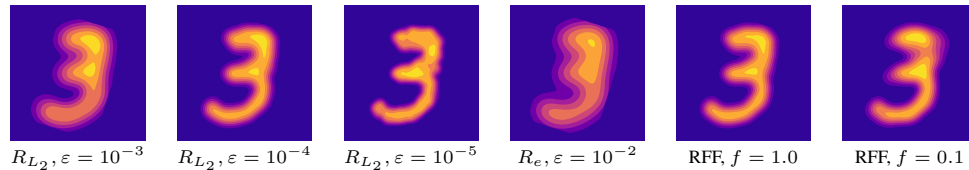

$R_{L_2}, \varepsilon = 10^{-3}$  $R_{L_2}, \varepsilon = 10^{-4}$  $R_{L_2}, \varepsilon = 10^{-5}$  $R_e, \varepsilon = 10^{-2}$  RFF, $f = 1.0$  RFF, $f = 0.1$

Figure 2: Comparison of regularization and parameterization choices. Labels at the bottom row are the regularizer type and the value of the constant $\varepsilon$ as in (4). $R_{L_2}, R_e$ means using quadratic and entropic regularization respectively. The last two columns show the result of using random Fourier features [RR08] instead of neural networks, with $f$ indicating the scale of the frequencies used.

The first three columns confirm that smaller $\varepsilon$ gives sharper results as the computed barycenter tends to the unregularized barycenter. On the other hand, entropic regularization yields a smoother marginal, but smaller $\varepsilon$ leads to numerical instability: we display the smallest one we could reach. The last two columns show that parameterization using random Fourier features [RR08] gives a comparable result as using neural networks, but the scale of the frequencies needs to be fine-tuned.

**Multivariate Gaussians with varying dimensions.** When the input distributions are multivariate Gaussians, the (unregularized) barycenter is also a multivariate Gaussian, and an efficient fixed-point algorithm can be used to recover its parameters [Álv+16]. We compute the ground truth barycenter of 5 randomly generated multivariate Gaussians in varying dimensions using [Álv+16] and compare our proposed algorithm to other state-of-the-art barycenter algorithms. Since measuring the Wasserstein distance of two distributions in high dimensions is computationally challenging, we instead compare the MLE parameters if we fit a Gaussian to the computed barycenter samples and compare with the true parameters. See Table 1 for the results of our algorithm with quadratic regularization compared with those from other state-of-the-art free-support methods. Among the Monge map estimation methods, the gradient-based Monge map (d) works the best in higher dimensions, and the result of (e) is slightly worse: we believe this is due to the error accumulated in the second stochastic gradient descent used to compute (14). For brevity, we only include (d) in Table 1. Note that

| Dimension | [CD14] | [CCS18] | Ours with (d) and $R_{L_2}$ |
|---|---|---|---|
| 2 | $\mathbf{7.28{\times}10^{-4}}(\mathbf{9.99{\times}10^{-5}})$ | $2.39{\times}10^{-3}(3.14{\times}10^{-4})$ | $1.98{\times}10^{-3}(1.17{\times}10^{-4})$ |
| 3 | $\mathbf{4.96{\times}10^{-3}}(\mathbf{6.42{\times}10^{-4}})$ | $8.97{\times}10^{-3}(9.22{\times}10^{-4})$ | $5.05{\times}10^{-3}(6.32{\times}10^{-4})$ |
| 4 | $1.35{\times}10^{-2}(1.73{\times}10^{-3})$ | $2.50{\times}10^{-2}(1.68{\times}10^{-3})$ | $\mathbf{1.22{\times}10^{-2}}(\mathbf{1.44{\times}10^{-3}})$ |
| 5 | $2.43{\times}10^{-2}(1.87{\times}10^{-3})$ | $5.05{\times}10^{-2}(2.22{\times}10^{-3})$ | $\mathbf{1.52{\times}10^{-2}}(\mathbf{1.18{\times}10^{-3}})$ |
| 6 | $4.38{\times}10^{-2}(2.04{\times}10^{-3})$ | $8.86{\times}10^{-2}(2.58{\times}10^{-3})$ | $\mathbf{2.37{\times}10^{-2}}(\mathbf{3.24{\times}10^{-3}})$ |
| 7 | $5.91{\times}10^{-2}(1.26{\times}10^{-3})$ | $1.24{\times}10^{-1}(1.63{\times}10^{-3})$ | $\mathbf{4.07{\times}10^{-2}}(\mathbf{2.65{\times}10^{-3}})$ |
| 8 | $8.31{\times}10^{-2}(1.23{\times}10^{-3})$ | $1.64{\times}10^{-1}(1.48{\times}10^{-3})$ | $\mathbf{4.23{\times}10^{-2}}(\mathbf{3.14{\times}10^{-3}})$ |

Table 1: Comparison of free-support barycenter algorithms on multivariate Gaussians of varying dimensions. Reported are the covariance difference $\|\Sigma - \Sigma^*\|_F$ where $\Sigma$ is the MLE covariance of the barycenter computed by each method, $\Sigma^*$ is the ground truth covariance, and $\|\cdot\|_F$ is the Frobenius norm. Smaller is better. All experiments are repeated 5 times with the mean and standard deviation reported. We use 5000 and 100 support points in [CD14] and [CCS18] respectively, as these are the maximum numbers allowed for the algorithms to terminate in a reasonable amount of time.

| [CD14] | [CCS18] | Ours with (d) and $R_{L_2}$ |
|---|---|---|
| $2.56{\times}10^{-7}(2.17{\times}10^{-9})$ | $9.37{\times}10^{-4}(4.84{\times}10^{-5})$ | $\mathbf{2.43{\times}10^{-7}}(\mathbf{6.57{\times}10^{-8}})$ |

Table 2: Comparison of subset posterior aggregation results in the covariance difference $\|\Sigma - \Sigma^*\|$, where $\Sigma$ is the covariance of the barycenter samples from each method, and $\Sigma^*$ is that of the full posterior. All experiments are repeated 20 times with the mean and standard deviation reported. As in Table 1, we use 5000 support points in [CD14] and 100 support points in [CCS18] as these are the maximum numbers permitted for the algorithms to terminate in a reasonable amount of time.

discrete fixed-support algorithms will have trouble scaling to higher dimensions as the total number of grid points grows exponentially with the number of dimensions. For instance, the covariance difference between the ground truth and those from running [Sta+17] with $10^5$ support points in $\mathbb{R}^4$ is $5.99 \times 10^{-2}(\pm 6.19 \times 10^{-3})$, which is significantly worse than the ones shown in Table 1. See the supplementary document for more details. In this experiment, we are able to consistently outperform state-of-the-art free-support methods in higher dimensions with the additional benefit of providing sample access from the barycenter.

**Subset posterior aggregation.** To show the effectiveness of our algorithm in real-world applications, we apply our method to aggregate subset posterior distributions using barycenters, which has been shown as an effective alternative to the full posterior in the massive data setting [Sri+15; SLD18; Sta+17]. We consider Poisson regression for the task of predicting the hourly number of bike rentals using features such as the day of the week and weather conditions.[3] We use one intercept and 8 regression coefficients for the Poisson model, and consider the posterior on the 8-dimensional regression coefficients. We randomly split the data into 5 equally-sized subsets and obtain $10^5$ samples from each subset posterior using the Stan library [Car+17].

The barycenter of subset posteriors converges to the full data posterior [SLD18]. Hence, to evaluate the quality of the barycenter computed from the subset posterior samples, we use the full posterior samples as the ground truth and report the differences in covariance using sufficiently many samples from the barycenter, and compare against other free-support barycenter algorithms (Table 2). See the supplementary document for more details. To show how the quality of the barycenter improves as more samples are used from our barycenter, we plot the 2-Wasserstein distance versus the number of samples in Figure 3. Since computing $W_2(\nu, \nu^*)$ requires solving a large linear program, we are only able to produce the result up to 15000 samples. This is also a limitation in [CD14] as each iteration of their alternating optimization solves many large linear programs; for this reason we are only able to use 5000 support points for their method. We see that as we use more samples, $W_2$ steadily decreases with lower variance, and we expect the decrease to continue with more samples. With 15000 samples our barycenter is closer to the full posterior than that of [CD14].

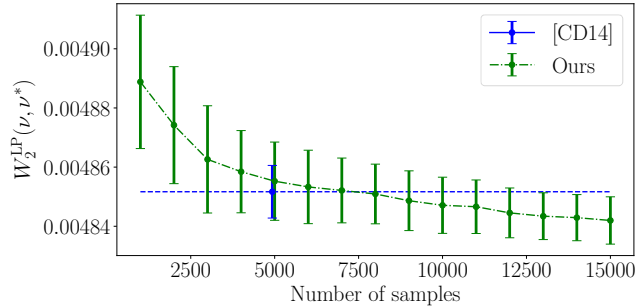

Figure 3: 2-Wasserstein distance versus the number of samples from the output of our algorithm with (d). Same number of points are used from both the full posterior and the computed barycenter to compute $W_2^{\mathrm{LP}}(\nu, \nu^*)$. The blue bar is the result of [CD14] with 5000 support points. The caps around each solid dot indicate the standard deviation across 20 independent trials.

## 5  Conclusion

Our stochastic algorithm computes the barycenter of continuous distributions without discretizing the output barycenter, and has been shown to provide a clear advantage over past methods in higher dimensions. However the performance of our algorithm still suffers from a curse of dimensionality when the dimension is too high. Indeed in this case the support measure we fix from the beginning becomes a poor proxy for the true barycenter, and an enormous batch size is required to evaluate the expectation in (11) with reasonably small variance. One future direction is to find a way to estimate the support measure dynamically, but choosing a representation for this task is challenging. Another issue that can be addressed is to reduce regularization bias. This can either happen by formulating alternative versions of the dual problem or by improving the methods for estimating a Monge map.

## Broader Impact

In this paper, we propose new techniques for estimating Wasserstein barycenters. The Wasserstein barycenter is a purely mathematical notion that can be applied to solve a variety of problems in practice. Beyond generic misuses of statistical and machine learning techniques, we are not aware of any unethical applications specifically of Wasserstein barycenters and hope that practitioners will find use cases for our methodology that can benefit society.

## Acknowledgments

Justin Solomon and the MIT Geometric Data Processing group acknowledge the generous support of Army Research Office grants W911NF1710068 and W911NF2010168, of Air Force Office of Scientific Research award FA9550-19-1-031, of National Science Foundation grant IIS-1838071, from the CSAIL Systems that Learn program, from the MIT–IBM Watson AI Laboratory, from the Toyota–CSAIL Joint Research Center, from a gift from Adobe Systems, and from the Skoltech–MIT Next Generation Program.

## Footnotes

[1]In some convention, there is an additional exponent in the summands of (8) such as $\sum_{i=1}^n \lambda_i W_2^2(\mu_i, \nu)$ for the 2-Wasserstein barycenter. Here we absorb such exponent in (1), e.g., $W(\mu_i, \nu) = W_2^2(\mu_i, \nu)$.

[2]We ran our experiments using a NVIDIA Tesla V100 GPU on a Google cloud instance with 12 compute-optimized CPUs and 64GB memory.

[3]http://archive.ics.uci.edu/ml/datasets/Bike+Sharing+Dataset

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
