[Supplementary Material]

# A  Proof of Theorem 3.1

We restate Theorem 3.1 below.

**Theorem A.1.** *The dual problem of*

$$\inf_{\nu \in \mathcal{M}_1^+(\mathcal{X})} \sum_{i=1}^n \lambda_i W_R^{\mu_i \otimes \eta}(\mu_i, \nu). \tag{A.1}$$

*is*

$$\sup_{\substack{\{(f_i, g_i)\}_{i=1}^n \subset C(\mathcal{X})^2 \\ \sum_{i=1}^n \lambda_i g_i = 0}} \sum_{i=1}^n \lambda_i \left( \int f_i \, \mathrm{d}\mu_i - \iint R^* \left( f_i(x) + g_i(y) - c(x,y) \right) \, \mathrm{d}\mu_i(x) \, \mathrm{d}\eta(y) \right). \tag{A.2}$$

*Moreover, strong duality holds in the sense that the infimum of* (A.1) *equals the supremum of* (A.2), *and a solution to* (A.1) *exists. If* $\{(f_i, g_i)\}_{i=1}^n$ *solves* (A.2), *then each* $(f_i, g_i)$ *is a solution to the dual formulation* (5) *of* $W_R^{\mu_i \otimes \eta}(\mu_i, \nu)$ *where* $\nu$ *is a solution to* (A.1). *That is,*

$$W_R^{\mu_i \otimes \eta}(\mu_i, \nu) = \sup_{f, g \in C(\mathcal{X})} \int f \mathrm{d}\mu + \int g \mathrm{d}\nu - \int R^* \left( f \oplus g - c \right) \mathrm{d}\mu_i \otimes \eta \tag{A.3}$$

$$= \int f_i \mathrm{d}\mu + \int g_i \mathrm{d}\nu - \int R^* \left( f_i \oplus g_i - c \right) \mathrm{d}\mu_i \otimes \eta,$$

*where we write* $(f \oplus g)(x, y) \triangleq f(x) + g(y)$.

*Proof.* We first prove the strong duality. We view $C(\mathcal{X})$ as a normed vector space with the supremum norm. Let $V = \oplus_{i=1}^{2n} C(\mathcal{X})$ be the direct sum vector space endowed with the natural norm, i.e., for $u = \{(f_i, g_i)\}_{i=1}^n \in V$,

$$\|u\| \triangleq \sum_{i=1}^n (\|f_i\| + \|g_i\|).$$

For brevity, denote $\xi_i \triangleq \mu_i \otimes \eta$. We use the notation $\xi_i$ to suggest that a more general support measure can be used to establish the strong duality. Define $J : V \to \mathbb{R}$ to be, for $u = \{(f_i, g_i)\}_{i=1}^n$,

$$J(u) \triangleq \sum_{i=1}^n \lambda_i \left( \iint R^*(f_i(x) + g_i(y) - c(x,y)) \mathrm{d}\xi_i(x,y) - \int f_i \mathrm{d}\mu_i \right)$$

$$= \sum_{i=1}^n \lambda_i \left( \iint R^*(f_i \oplus g_i - c) \mathrm{d}\xi_i - \int f_i \mathrm{d}\mu_i \right). \tag{A.4}$$

Let $Y \triangleq C(\mathcal{X})$. By the Riesz–Markov–Kakutani representation theorem, the continuous dual space of $Y$ is $Y^* = \mathcal{M}(\mathcal{X})$ with the pairing $\langle p^*, p \rangle = \int p \mathrm{d}p^*$ for $p^* \in Y^*, p \in Y$. Define $B : V \to Y$ as, for $u = \{(f_i, g_i)\}_{i=1}^n$,

$$B(u) = B(\{(f_i, g_i)\}_{i=1}^n) \triangleq - \sum_{i=1}^n \lambda_i g_i.$$

Then the negative of (A.2) becomes

$$\inf_{\substack{u \in V \\ B(u)=0}} J(u),$$

where the equality $B(u) = 0$ is component-wise (i.e. $B(u)$ is the constant-zero function in $Y$). Similarly we use $\leq, <$ to mean component-wise inequality in $C(\mathcal{X})$. Since $R^* : \mathbb{R} \to \mathbb{R}$ is increasing in our assumption for quadratic and entropic regularization (6), the above program is the same as

$$\inf_{\substack{u \in V \\ B(u) \leq 0}} J(u). \tag{A.5}$$

This is because if $u \in \{u|B(u) \le 0\} \setminus \{u|B(u) = 0\}$, then for some $x \in \mathcal{X}$, $\sum_{i=1}^n \lambda_i g_i(x) > 0$, and by replacing every $g_i$ with $g_i - \sum_{i=1}^n \lambda_i g_i(x)$ the objective (A.5) can only gets smaller.

The dual problem of (A.5) can be calculated as (see Chapter III.(5.23) in [ET99])

$$\sup_{\nu \le 0} \inf_{u \in V} \left\{ -\int B(u)\mathrm{d}\nu + J(u) \right\}$$

$$= \sup_{\nu \le 0} \inf_{\{(f_i, g_i)\}_{i=1}^n \subset C(\mathcal{X})^2} -\int \left( -\sum_{i=1}^n \lambda_i g_i \right) \mathrm{d}\nu + \sum_{i=1}^n \lambda_i \left( \iint R^*(f_i \oplus g_i - c)\mathrm{d}\xi_i - \int f_i \mathrm{d}\mu_i \right)$$

$$= \sup_{\nu \ge 0} \inf_{\{(f_i, g_i)\}_{i=1}^n \subset C(\mathcal{X})^2} \sum_{i=1}^n \lambda_i \left( \iint R^*(f_i \oplus g_i - c)\mathrm{d}\xi_i - \int f_i \mathrm{d}\mu_i - \int g_i \mathrm{d}\nu \right)$$

$$= \sup_{\nu \ge 0} \sum_{i=1}^n \lambda_i \inf_{(f_i, g_i) \in C(\mathcal{X})^2} \left( \iint R^*(f_i \oplus g_i - c)\mathrm{d}\xi_i - \int f_i \mathrm{d}\mu_i - \int g_i \mathrm{d}\nu \right) \tag{A.6}$$

$$= \sup_{\nu \ge 0} \sum_{i=1}^n -\lambda_i W_R^{\xi_i}(\mu_i, \nu) \tag{A.7}$$

$$= - \inf_{\nu \ge 0} \sum_{i=1}^n \lambda_i W_R^{\xi_i}(\mu_i, \nu). \tag{A.8}$$

To get (A.7) we used the duality for regularized Wasserstein distance (5).

In order to apply classical results from convex analysis (for instance, Proposition 5.1 of Chapter III in [ET99]) to establish strong duality and the existence of solutions, we need to show:

(a)  $J$ is a convex l.s.c. (lower-semicontinuous) function.

(b)  $B$ is convex with respect to $\le$.

(c)  For any $\nu \in Y^*$, $\nu \ge 0$, the map $u \mapsto \int B(u)\mathrm{d}\nu$ is l.s.c..

(d)  $\{u \in V | B(u) \le 0\} \ne \varnothing$.

(e)  There exists $u_0 \in V$ such that $-B(u_0) < 0$.

(f)  The infimum in (A.5) is finite.

Since $B$ is linear and $Y = C(\mathcal{X})$ in our case, the conditions (b)-(e) are satisfied automatically. Convexity of $J$ (a) follows because $R^*$ is convex so that, for $u_j = \{(f_i^{(j)}, g_i^{(j)})\}_{i=1}^n$, $j \in \{1, 2\}$, and $\theta \in [0, 1]$,

$$J(\theta u_1 + (1 - \theta)u_2)$$

$$= \sum_{i=1}^n \lambda_i \left( \iint R^*((\theta f_i^{(1)} + (1 - \theta)f_i^{(2)}) \oplus \theta(g_i^{(1)} + (1 - \theta)g_i^{(2)})) - c)\mathrm{d}\xi_i \right.$$

$$\left. - \int (\theta f_i^{(1)} + (1 - \theta)f_i^{(2)})\mathrm{d}\mu_i \right)$$

$$\le \theta \sum_{i=1}^n \lambda_i \left( \iint R^*(f_i^{(1)} \oplus g_i^{(1)} - c)\mathrm{d}\xi_i - \int f_i^{(1)}\mathrm{d}\mu_i \right)$$

$$+ (1 - \theta) \sum_{i=1}^n \lambda_i \left( \iint R^*(f_i^{(2)} + g_i^{(2)} - c)\mathrm{d}\xi_i - \int f_i^{(2)}\mathrm{d}\mu_i \right)$$

$$= \theta J(u_1) + (1 - \theta)J(u_2).$$

Next we show that $J$ is l.s.c. with respect to the norm topology on $V$. Since $J$ is convex and does not take on values $\pm\infty$, by Proposition III.2.5 of [ET99], it is enough to show that $J$ is bounded above in a neighborhood of 0. Fix any $\delta > 0$. As before we write $u = \{(f_i, g_i)\}_{i=1}^n \in V$. Then $\|u\| < \delta$

implies $\sup_{x\in\mathcal{X}}\max(f_i(x),g_i(x))<\delta$ for all $i$. Since $\mathcal{X}$ is compact, $\sup_{x,y\in\mathcal{X}}c(x,y)$ is bounded. Hence the integrand in (A.4) is bounded for $\|u\|<\delta$ as $R^*$ is increasing, and the conclusion that $J$ is bounded on $\{u\in V|\|u\|<\delta\}$ follows from the fact that both $\xi_i$ and $\mu_i$ are probability measures for all $i$. This proves $J$ is continuous, and hence l.s.c..

It remains to show the infimum in (A.5) is finite. Note that for $u\in V$ such that $B(u)\le 0$, we have $\sum_{i=1}^n \lambda_i g_i \ge 0$. Hence in this case, if like before we denote $\eta$ the uniform measure on $\mathcal{X}$, then

$$
\begin{aligned}
J(u) &= \sum_{i=1}^n \lambda_i \left( \iint R^*(f_i\oplus g_i - c)\mathrm{d}\xi_i - \int f_i\mathrm{d}\mu_i \right) \\
&\ge \sum_{i=1}^n \lambda_i \left( \iint R^*(f_i\oplus g_i - c)\mathrm{d}\xi_i - \int f_i\mathrm{d}\mu_i \right) - \int \left( \sum_{i=1}^n \lambda_i g_i \right)\mathrm{d}\eta \\
&= \sum_{i=1}^n \lambda_i \left( \iint R^*(f_i\oplus g_i - c)\mathrm{d}\xi_i - \int f_i\mathrm{d}\mu_i - \int g_i\mathrm{d}\eta \right) \\
&\ge -\sum_{i=1}^n \lambda_i W_R^{\mu_i\otimes\eta}(\mu_i,\eta) \\
&> -\infty.
\end{aligned}
$$

Thus by Proposition III.5.1 of [ET99], the problem (A.5) is stable (Definition III.2.2 in [ET99]), and in particular normal, so we have strong duality (Proposition III.2.1, III.2.2 in [ET99]), and the dual problem (A.8) has at least one solution. We comment that this does not imply (A.5) has a solution.

To show the solution $\nu^*$ to (A.8) is actually a probability measure, suppose $\nu^*(\mathcal{X})\ne 1$. Consider the inner infimum in (A.6) for a particular $i$. For any $t\in\mathbb{R}$, we can set $f=t$ and $g=-t$. Then

$$
\begin{aligned}
&\int R^*\left( f\oplus g - c) \right)\mathrm{d}\xi_i - \int f\mathrm{d}\mu_i - \int g\mathrm{d}\nu^* \\
&= \int R^*\left(-c)\right)\mathrm{d}\xi_i + t(\nu^*(\mathcal{X}) - \mu_i(\mathcal{X})) \\
&\le R^*(0) + t(\nu^*(\mathcal{X}) - \mu_i(\mathcal{X})),
\end{aligned}
$$

where we used the fact that $R^*$ is increasing and $c\ge 0$. Either sending $t\to\infty$ or $t\to -\infty$ shows that the minimizer $\nu^*$ must satisfy $\nu^*(\mathcal{X}) = \mu_i(\mathcal{X}) = 1$, for otherwise the infimum would be $-\infty$, which contradicts strong duality and (f).

Finally we prove the last statement of Theorem A.1. That is, if $\{(f_i,g_i)\}_{i=1}^n$ solves (A.2), then each pair $(f_i,g_i)$ solves (A.3). Suppose that $\{(f_i,g_i)\}_{i=1}^n$ solves (A.2). Let $\nu^*$ denote the solution to (A.1). Then $\sum_{i=1}^n \lambda_i g_i = 0$. So the supremum of (A.2) equals

$$
\begin{aligned}
&\sum_{i=1}^n \lambda_i \left( \int f_i\mathrm{d}\mu_i + \int g_i\mathrm{d}\nu^* - \iint R^*(f_i\oplus g_i - c)\mathrm{d}\mu_i\mathrm{d}\eta \right) \\
&\le \sum_{i=1}^n \lambda_i W_R^{\mu_i\otimes\eta}(\mu_i,\nu^*),
\end{aligned}
\tag{A.9}
$$

where the inequality follows from the duality (A.3) of the regularized Wasserstein distance. By strong duality we just showed, the supremum of (A.2) equals the infimum (A.1) which is (A.9). Hence inequality in (A.9) is equality, and we see that each pair $(f_i,g_i)$ solves (A.3). $\qquad\square$

## B  Experimental details and additional results

**Multivariate Gaussians with varying dimensions.** We generate the multivariate Gaussians in dimension $d$ used in Table 1 in the following manner. The mean is chosen uniformly at random in $[-1,1]^d$. The covariance matrix is obtained by first sampling a matrix $A$ with uniform entries in $[-0.3,0.3]$ and then taking $AA^\top$ as the covariance matrix. We reject $A$ if its condition number (computed with respect to 2-norm) is not in $[2,80]$.

We show in Table B.1 additional results for our algorithm with different choices of Monge map estimation methods and regularizers; in the last column we show the result of [CD14] where we use Sinkhorn algorithm [Cut13] instead of LP (see Table 1 for results with LP) to obtain the transport plan at every iteration.

| $d$ | Ours with (d) and $R_{L_2}$ | Ours with (e) and $R_{L_2}$ | Ours with (d) and $R_e$ | [CD14] with Sinkhorn |
|---|---|---|---|---|
| 2 | $\mathbf{1.98{\times}10^{-3}(1.17{\times}10^{-4})}$ | $2.38{\times}10^{-3}(2.48{\times}10^{-4})$ | $8.25\times10^{-3}(5.02\times10^{-4})$ | $5.22\times10^{-2}(5.09\times10^{-4})$ |
| 3 | $\mathbf{5.05{\times}10^{-3}(6.32{\times}10^{-4})}$ | $5.70{\times}10^{-3}(6.90{\times}10^{-4})$ | $8.15\times10^{-3}(6.50\times10^{-4})$ | $7.46\times10^{-2}(3.87\times10^{-4})$ |
| 4 | $\mathbf{1.22{\times}10^{-2}(1.44{\times}10^{-3})}$ | $1.27{\times}10^{-2}(1.19{\times}10^{-3})$ | $2.06\times10^{-2}(7.40\times10^{-4})$ | $8.78\times10^{-2}(1.40\times10^{-3})$ |
| 5 | $\mathbf{1.52{\times}10^{-2}(1.18{\times}10^{-3})}$ | $2.33{\times}10^{-2}(2.86{\times}10^{-3})$ | $3.72\times10^{-2}(9.81\times10^{-4})$ | $1.00\times10^{-1}(7.30\times10^{-4})$ |
| 6 | $\mathbf{2.37{\times}10^{-2}(3.24{\times}10^{-3})}$ | $3.27{\times}10^{-2}(2.63{\times}10^{-3})$ | $6.13\times10^{-2}(2.69\times10^{-3})$ | $1.10\times10^{-1}(7.93\times10^{-4})$ |
| 7 | $\mathbf{4.07{\times}10^{-2}(2.65{\times}10^{-3})}$ | $4.83{\times}10^{-2}(2.90{\times}10^{-3})$ | $8.42\times10^{-2}(4.62\times10^{-4})$ | $1.16\times10^{-1}(5.44\times10^{-4})$ |
| 8 | $\mathbf{4.23{\times}10^{-2}(3.14{\times}10^{-3})}$ | $4.79{\times}10^{-2}(2.46{\times}10^{-3})$ | $1.20\times10^{-1}(2.38\times10^{-3})$ | $1.18\times10^{-1}(7.07\times10^{-4})$ |

Table B.1: Additional results for the multivariate Gaussian experiment. Reported are the covariance difference $\|\Sigma - \Sigma^*\|_F$ where $\Sigma$ is the MLE covariance of the barycenter computed by each method, $\Sigma^*$ is the ground truth covariance, and $\|\cdot\|_F$ is the Frobenius norm. Smaller is better. All experiments are repeated 5 times with the mean and standard deviation reported. Here $R_{L_2}$ refers to quadratic regularization with $\varepsilon = 10^{-4}$, and $R_e$ refers to entropic regularization with $\varepsilon = 0.1$. The regularizing $\varepsilon$ is further scaled with respect to the diagonal length of the bounding box squared. For [CD14] with Sinkhorn algorithm, we choose $\varepsilon = 0.1$.

To briefly comment on the runtime of our algorithm (with (d)) and that of [CD14] and [CCS18], in the 8-dimensional Gaussian experiment from Table 1, our algorithm takes around 15 minutes, [CD14] takes 20 minutes, while [CCS18] takes an hour or longer. The simple form of Algorithm 1 and the convex nature of (11) give rise to fast convergence of our approach.

**Subset posterior aggregation.** We adopted the `BikeTrips` dataset and preprocessing steps from `https://github.com/trevorcampbell/bayesian-coresets` [CB19]. The posterior samples in the subset posterior aggregation experiment are generated using NUTS sampler [HG14] implemented by the Stan library [Car+17]. To enforce appropriate scaling of the prior in the subset posteriors we use stochastic approximation trick [SLD18], i.e. scaling the log-likelihood by the number of subsets. Please see the code for further details.

In Table B.2, we show additional results comparing [CD14] and our algorithm in three different losses: difference in mean, covariance, and the (unregularized) 2-Wasserstein distance computed using 5000 samples. See Figure 3 for a comparison with varying number of samples used to compute the 2-Wasserstein distance.

| Loss | [CD14] | Ours with (d) and $R_{L_2}$ | Ours with (e) and $R_{L_2}$ |
|---|---|---|---|
| $\|\mu - \mu^*\|$ | $4.79\times10^{-3}(3.19\times10^{-6})$ | $4.79\times10^{-3}(5.96\times10^{-7})$ | $\mathbf{4.79{\times}10^{-3}(1.80{\times}10^{-7})}$ |
| $\|\Sigma - \Sigma^*\|$ | $2.56\times10^{-7}(2.17\times10^{-9})$ | $\mathbf{2.43{\times}10^{-7}(6.57{\times}10^{-8})}$ | $9.51\times10^{-7}(6.62\times10^{-9})$ |
| $W_2^{\mathrm{LP}}(\nu, \nu^*)$ | $\mathbf{4.85{\times}10^{-3}(8.90{\times}10^{-6})}$ | $4.86\times10^{-3}(9.40\times10^{-6})$ | $4.96\times10^{-3}(6.10\times10^{-6})$ |

Table B.2: Comparison of subset posterior aggregation results in difference in mean, covariance, and 2-Wasserstein distance. All experiments are repeated 20 times with the mean and standard deviation reported. Variables with a superscript star ($\mu^*, \Sigma^*, \nu^*$) are quantities from the full posterior, and variables without a star are from the computed barycenter. The mean and covariance are estimated with sufficiently many samples from the barycenter, while the 2-Wasserstein distance is computed using 5000 samples from both the barycenter and the full posterior.