[Reviews · NeurIPS 2020]

Review 1

Summary and Contributions: This paper proposes a new method for computing the Wasserstein barycenter. The approach consists of solving the dual of a regularized version of the original barycenter problem by stochastic gradient algorithm and then recovering the desired barycenter by some complementarity-type conditions, thanks to the strong duality. The main theoretical contribution is Theorem 3.1 which shows dual problem of the regularized barycenter problem and that the strong duality holds. The numerical section is also nice: experiments based on both synthetic and real data-sets are presented, with clear discussions on the implementation details and explanations on the experimental results.

Strengths: i) the support of the barycenter needs not be fixed a priori and can even be continuous; ii) samples of the barycenter comes with the barycenter without additional computational cost; iii) The numerical section is good for reasons described as above.

Weaknesses: The procedure for recovering the barycenter from the dual solution (described in Section 3.3) is computational costly and lacks theoretical analysis.

Correctness: As far as I could verify, both the theoretical results and the empirical methodology are correct.

Clarity: The paper is generally quite well written, with only a few minor typographical mistakes.

Relation to Prior Work: Although the discussion spent on prior works is quite thin in view of the vast literature on Wasserstein barycenter, the comparison to the most related works is clear and rather adequate.

Reproducibility: Yes

Additional Feedback: The references contain a lot of "et al". I don't know if it was intentional or not. But this is not a good and not a usual practice in the community. UPDATE: --------------------------------------------- It has been pointed out by the other reviewers that the main result Theorem 3.1 (and its proof technique) is not new and similar to a number of existing results. Also, my concerns are not addressed at all. So, I decided to lower my score. ---------------------------------------------


Review 2

Summary and Contributions: This paper proposes a method for the computing the regularized Wasserstein barycenter between discrete measures without discretizing the support of the barycenter. The dual problem is considered and solved with a stochastic gradient descent. Once optimal dual variables have been obtained, different strategies are proposed to compute the barycenter on any given grid. Most of them relies on the approximation of the Monge map between discrete densities and the (continuous) baycenter.

Strengths: Computing Wasserstein Barycenter without discretizing the support is a very relevant challenge when high dimensions are involved. Considering the dual problem is a nice way to do so. This is an intresting problem for the community.

Weaknesses: - The main contribution is to propose 5 strategies to recover the barycenter. There is no real conclusion on what one should do in practice (at least pro and cons). The authors suggest to consider strategy (d), but as mentionned in the paper, this is a method for unregularized transport. - The ideas and the implementation strategies can already be found in the works of Cuturi, Genevay, Seguy... For the final barycenter estimation, the main novelty is the MCMC sampling strategy, which is not very efficient. The remaining ones are existing approaches coming from other papers. - As the method requires an acceptable discretized proxy for the barycenter, it does not really get rid of the original dimensional issue. Moreover, estimating the Monge map is itself a very complex problem. - In the experiments, the proposed approach does not lead to significant improvements in higher dimensions (Tables and Figure 3). The method of Salomon can be very fast to estimate regularized barycenters with quadratic cost. A discussion on the overall computational cost (including the estimation of the dual variables and the different strategies) shoud also be added. The influence of the proxy could have been illustrated.

Correctness: - I did not check the whole proof of the main theorem which is technical. A similar result can be found in the paper of Peyré and Cuturi: A Smoothed Dual Approach for Variational Wasserstein Problems, SIAM SIMMS 2015. - The way the linear constraint is removed to get an unconstrained problem (eq. (10)->(11)) is not standard for me. A reference or a discussion should be added.

Clarity: The paper is well written and easy to read.

Relation to Prior Work: The references are adequate. Other works could have been mentionned, but the recent literature is so large that I can't blame the authors.

Reproducibility: Yes

Additional Feedback:


Review 3

Summary and Contributions: The paper proposes algorithms to compute barycenters of continuous probability distributions with respect to regularized Wasserstein distances. One contribution of the paper is to derive the dual problem of the regularized Barycenter task in function space, by applying a general Fenchel-Rockafellar theorem. Then, different approximating spaces (either neural networks or RKHS) for the dual problem are considered. The finite-dimensional problem is solved in an online fashion using stochastic optimization, requiring only samples from the distributions. Another contribution is the detailed evaluation of the discretizations, and particular care is taken towards recovery of the primal Barycenter solution.

Strengths: The proposed method based on stochastic optimization, regularization, and dual approximation with neural nets can be seen as a non-standard approach to continuous variational problems. I find it to be quite interesting and the abstract approach can be impactful beyond the Wasserstein barycenter problem. I really liked the comparison of different primal recovery strategies. For NeurIPS, the topic is a bit niche and the results/methods might eventually be of much greater interest to the graphics, imaging, and scientific computing communities as a potentially scalable alternative to finite elements.

Weaknesses: The main weakness I see with this work is the fact that it pursues a specific (and interesting!) solution strategy for continuous variational problems, but unfortunately while missing a bit the "big/abstract picture". For such variational problems, a standard and well-developed/established solution method with strong theoretical guarantees (e.g. convergence rates) exist: finite element methods. It should be at least mentioned in the paper why the existing literature on finite element approaches (also for the Wasserstein barycenter problem) is a bit ignored. In fact, the RKHS discretization could perhaps be connected to such strategies, as finite elements can be seen as a finite-dimensional RKHS with the finite element basis functions as a feature map. An approximation of the dual problem is considered, but this choice is only motivated perhaps a bit superficially: of course, together with the regularization, it makes the problem amenable to online/stochastic optimization. However, there are more intricate aspects, which could be worth discussing: the dual approximation leads to a lower bound of the original objective. By a restriction of the dual variables (test functions), one loses the ability to distinguish the primal variables (measures). Intuitively, if one restricts the ability to test, one gets worse at distinguishing. Formally, the dual approximation induces a quotient space of measures in the primal which can lead to a (perhaps even desirable?) superposition of primal solutions. Interestingly, the strongly convex regularizer could help there, to enforce a unique candidate from the equivalence class. A different strategy would be to perform an approximation in the primal (i.e., considering a finite-dimensional space of measures). This would lead to an upper bound of the original objective instead of a lower-bound. The paper misses to discuss the benefits of dual approximation vs. primal approximation, and their interplay with regularization. As there is no free lunch, each of the strategies comes with its advantages and disadvantages. For instance, while the regularize + dual strategy allows easy online optimization, one struggles heavily with a good recovery of the primal solution (which is however adequately addressed in the paper).

Correctness: All theoretical results are derived in a clean and rigorous fashion.

Clarity: The paper is very well written and was easy and clear to read.

Relation to Prior Work: The paper prominently mentions [Gen+16] and [Seg+17], but perhaps the novelty regarding these two papers should be stated more clearly. While a different problem is studied (sum of Wasserstein distances instead of a single Wasserstein distance), the dualization, discretization, and optimization approaches are quite similar.

Reproducibility: Yes

Additional Feedback: What I like about the paper, is that the proposed regularization + stochastic optimization approach might yield scalable methods not only for the continuous Wasserstein Barycenter problem but eventually lead to new efficient solution methodologies for other variational problems posed on measures. Minor suggestions / ideas: - While clear what is meant, perhaps one could clarify the domain of R(t) in (4) so that the conjugate in Eq. (6) matches. - For the neural network discretization, recent insights for representation of low-dimensional functions (e.g., positional encodings, SIREN activations or Fourier features networks, see e.g. https://people.eecs.berkeley.edu/~bmild/fourfeat/) might greatly improve the performance over the simple fully connected approach. In particular, I have hope that it might remove some of the smaller artifacts in Figure 1. - Perhaps one could look into gamma-convergence of the regularized functional to the original one (given vanishing regularization parameter). This is standard for smoothing approaches to variational problems (e.g. the Ambrosio-Tortorelli approaches for the Mumford-Shah functional). - The approximation of dual variables for infinite-dimensional variational problems on measures has been considered in different contexts, for example in: [Numerical solution of soap film dual problems, Brakke '95], [Duality and the continuous graphical model, Fix & Agarwal, ECCV '14], [Sublabel-accurate discretization of nonconvex free-discontinuity problems, Moellenhoff & Cremers, ICCV '17]. Perhaps such variational problems could also benefit/be made more scalable by a similar strategy. == Update == After reading the rebuttal and discussions, I still think this is a solid paper and therefore keep my score. I encourage the authors to add paragraph on the relation to finite-element approaches to the final version.


Review 4

Summary and Contributions: The paper introduces an approach for computing 2-Wasserstein barycenters without discretization, relying on stochastic gradient descent based on Monte Carlo - type sampling. The approach considers the use of regularized (entropic and quadratic) 2-Wasserstein metric in the barycentric expression, whose dual problem the authors derive. Solving the dual problem yields dual potentials, that are also shown to relate to individual Kantorovich potentials between the barycenter and a measure in the population considered. These potentials are then used to sample from the barycenter by a considerable amount of different methods (5), consisting of integration strategies (a-b) and strategies based on approximating the Monge map (c-e). Finally, the method is illustrated in small-scale experiments, by comparing the different strategies (a-e), and comparing the best strategy to two other methods in the literature.

Strengths: The contribution is novel, and I personally like Theorem 3.1, specifically how the potentials solving (10) are the regularized Kantorovich potentials of given sub-problems. The authors consider a significant amount of approaches to solve the barycenter, given the potentials, which are well-explained. This is definitely a contribution to computing Wasserstein barycenters.

Weaknesses: The experimental section is sufficient, although personally I would have liked to see whether the MLPs used to approximate the dual potentials are sufficient, or whether the results could have been improved by considering a deeper/wider architecture. Furthermore, a comparison of computation times between the proposed method and the SOTA methods would have been welcome.

Correctness: I did not find any errors in the paper.

Clarity: The paper is well-written and clear, making it easy to follow. T

Relation to Prior Work: The paper has a section for related work, where it is clearly explained how the work builds on prior work.

Reproducibility: Yes

Additional Feedback: —DETAILED COMMENTS— - Table 1: Are the values in the parentheses the standard deviations? It might be a bit cleared by using the plus-minus sign. - Figure 3: For a better comparison, I think you should show the results for [CD14] for number of samples <= 5000, not just when =5000. - Figure 1: It might be easier to compare the results here if all the methods were visualized using a density map / samples. Now (a) is standing out a bit. On the other hand, comparison is still possible as it is. ## Post Rebuttal I would like to thank the authors for the rebuttal and for addressing my concerns. The added results would perhaps have raised my score, but I agree with R2 that some of the contributions of the paper are quite incremental. Therefore, I will be keeping my original score.

[Author Response · NeurIPS 2020]

We thank the reviewers for their time and their valuable feedback and thoughtful suggestions. We remain confident that our work is of strong interest to the NeurIPS community and can be further strengthened by incorporating the suggested changes in a final version of the paper. Below are our answers to the reviewers' comments, grouped by topic.

**Novelty of dual formulation. R2** considered our method requiring a "discretized proxy." This is not correct. The proxy support measure in the formulation is a continuous measure, and in the implementation we choose a uniform measure on the bounding box containing the support of all source distributions (estimated by samples). Such choice of support measure allows us to design a novel barycenter algorithm in a *free-support* and *continuous* setting: both the source distributions and the resulting barycenter are continuous (except for optional recovery method (a), but we suggest continuous alternatives). To the best of our knowledge, all prior works either consider *fixed-support* (e.g., [3]) or *discrete* source measures (e.g., [2]). We thank **R2** for pointing out [3] which we will add to the references. The method in [3] utilizes entropic regularization and uses a fixed discrete proxy, making it sufficient to parameterize only one of the dual potentials as the other can be retrieved in closed form based on first-order optimality [3]. For a continuous setting, this becomes infeasible as the closed form will become an integral in high dimensions. Moreover, such closed form is not possible for quadratic regularization which we found to be more stable and more accurate in higher dimensions.

**R2** and **R3** asked about the novelty of our approach over existing continuous methods on regularized OT distances [4, 5]. First of all, a different, more challenging optimization problem is studied in our work. The variables in the barycenter problem we consider include not only the individual transport plan from each source to the barycenter, but importantly also the barycenter itself. There is no clear way to extend [4, 5] to the barycenter setting without introducing non-convex min-max optimization. By introducing a novel regularizing measure $\mu_i \otimes \eta$ that does not rely on the unknown barycenter but only on a proxy measure $\eta$, we are able to encode the information of the barycenter in the dual potentials themselves without an explicit parametrization, thanks to Theorem 3.1. We will further emphasize the novelty of our method compared to [4, 5] in the final version of the related-work section.

**Comparison of recovery methods.** We agree with **R3** that the numerical comparison of 5 barycenter recovery methods is another highlight of the paper; even for computing regularized transport distance such comparison is lacking in the existing literature. In particular, we proposed the MCMC recovery strategy (b) which is suitable theoretically since we use continuous parametrization of the dual potentials (so gradient-based MCMC methods are viable) but is slow in practice. To address **R2**'s concern of lack of conclusion on which method to choose, we have included a discussion explaining each method's advantages and disadvantages in the paragraph "qualitative results in 2 and 3 dimensions" on page 6. We will add recommendation for higher-dimensional situations to accompany the results in Table B.1 and Table B.2. To address **R1**'s concern about recovery methods being costly, method (d) in fact comes at almost no cost (gradient computation), and method (e) requires a secondary SGD optimization that is no more costly than Algorithm 1.

**Relevance to NeurIPS. R3** suggested applying similar strategies to a wider range of variational problems of interest to NeurIPS. We thank **R3** for the detailed references, which we will incorporate into the related-work section. We would like to point out that there are three accepted papers at NeurIPS last year inspired by Wasserstein barycenters. The application to large-scale Bayesian posterior computation is also of considerable interest to the NeurIPS community.

**Theoretical analysis. R3** suggested analyzing the duality gap for the algorithm in practice since parametrization will restrict the functional space. On a related note, **R1** asked about theoretical analysis of recovery methods. These are challenging questions that depend on the specific structure of parameterization and the particular recovery method. Even for continuous regularized OT distance [5] this is not well-understood. We agree these are areas for future work.

**Experiments and improvements over SOTA. R2** is concerned about the level of improvements over SOTA. Our improvements are quite significant in the experiments: in Table 1, the covariance difference computed by our algorithm is consistently 1.5-2× smaller in higher dimensions as those of SOTA. Additionally, this example only deals with Gaussians, whose low sample complexity favors discretized methods more than other settings. **R2** mentions the efficiency of [6], but [6] is not applicable in high dimensions as a discretized grid is needed. While improvements in Table 2 appear more modest, covariance difference does not fully characterize discrepancy of measures for non-Gaussians. The comparison using 2-Wasserstein distance in Figure 3 reveals significant improvement, which we expect to further increase if we use more samples from our barycenter for the 2-Wasserstein distance. **R4** suggested testing with different MLP architectures. We tested with deeper/wider networks but found no noticeable improvement over the proposed architecture. We will add more details on the runtimes for different methods as **R4** suggested. In high dimensional experiments, our method takes around 15 minutes, [2] takes 20 minutes, and [1] takes an hour or longer. The relative ease of training is likely due to the convexity of our formulation (Equation 11).

[1] Claici et al. Stochastic Wasserstein barycenters. *Proc. ICML*, 2018.
[2] Cuturi & Doucet. Fast computation of Wasserstein barycenters. *J. Machine Learning Research*, 2014.
[3] Cuturi & Peyré. A smoothed dual approach for variational Wasserstein problems. *SIAM J. Imaging Sciences*, 2016.
[4] Genevay et al. Stochastic optimization for large-scale optimal transport. *Proc. NeurIPS*, 3440–3448, 2016.
[5] Seguy et al. Large-scale optimal transport and mapping estimation. *Proc. ICLR*, 2018.


[Meta-Review · NeurIPS 2020]

Three reviewers have positive opinions of the paper. R2 has pointed out that Theorem 3.1 is similar to others in the literature, and R3 observed that the theorem is proved via a more-or-less standard application of the Fenchel-Rockafellar duality theorem. Still, the paper is above the bar for acceptance.